# Frailty and 12-month mortality among older adults with type 2 diabetes in nursing homes: A longitudinal study

Maturin Tabué-Teguo[1,2], Nadine Simo[1,2], Axiane Placide-Francil[3], Moustapha Dramé [ID][1,2], Laurys Letchimy[1,2], Denis Boucaud-Maitre [ID][2,4]*

1 Centre Hospitalo-Universitaire de Martinique, Fort-de-France, Martinique, 2 Equipe EPICLIV, Université des Antilles, Fort-de-France, Martinique, 3 Université des Antilles, Fort-de-France, Martinique, 4 Centre Hospitalier le Vinatier, Bron, France

* denis.boucaud@gmail.com

## Abstract

### Background

Frailty is highly prevalent among older adults with type 2 diabetes mellitus (T2DM) and may contribute to adverse health outcomes, particularly in institutionalized settings. Despite its clinical relevance, the prognostic value of frailty among nursing home residents with T2DM remains underexplored. This study aimed to assess the association between frailty, assessed using the Frailty Index (FI), and 12-month all-cause mortality among older adults with T2DM residing in French Caribbean nursing homes.

### Methods

Data were drawn from the KASEHPAD (Karukera Study on Aging in Nursing Homes) study, a prospective, longitudinal cohort conducted across six nursing homes in Martinique and Guadeloupe. Frailty was assessed at baseline using a 30-item deficit accumulation model to compute the FI (range: 0–1). Mortality data were collected over a 12-month follow-up period. Associations between FI and mortality were analyzed using logistic regression and Cox proportional hazards models.

### Results

The study included 94 participants with T2DM (mean age: 81.1 ± 10.0 years; 42.6% male). The mean FI was 0.30 ± 0.14. Over the 12-month follow-up, 28 participants (29.8%) died. In unadjusted logistic regression models, each 0.01-point increase in FI was associated with a 6% increase in the odds of death (Odd Ratio (OR) = 1.06; 95% CI: 1.02–1.11; $p = 0.002$). After adjusting for age and sex, frailty was marginally associated with 1-year mortality (OR = 1.05; 95% CI: 1.00–1.10; $p = 0.056$), but was

**Data availability statement:** All relevant data are within the manuscript and its Supporting Information files.

**Funding:** This study was supported by a grant from the Conseil Départemental de la Guadeloupe and ARS de la Guadeloupe, Saint-Martin, and Saint-Barthélemy (grant 2020/DPAPH/DRM) and ARS Martinique. The funding body had no role in the design of the study and collection, analysis, and interpretation of data and in writing the manuscript.

**Competing interests:** The authors have declared that no competing interests exist.

not significantly associated with time to death in the Cox model (Hazard Ratio [HR] = 1.03; 95% CI: 0.99–1.07; p = 0.139).

## Conclusion

Frailty measured by the FI showed a tendency to be associated with short-term mortality among older adults with type T2DM living in nursing homes. These findings underscore the need for larger studies to further assess the prognostic utility of the FI in informing care planning and clinical management in this vulnerable population.

## Introduction

Type 2 diabetes mellitus (T2DM) is highly prevalent among older adults and is frequently associated with multiple comorbidities [1], functional decline, increased frailty measurement and elevated mortality risk [2,3]. In France, data from the GERODIAB cohort—a longitudinal study of nearly 1,000 individuals aged 70 years and older—have highlighted the frequent occurrence of diabetes-related complications, including cardiovascular events, renal impairment, cognitive decline, and high mortality over a five-year follow-up period [4]. In long-term care settings, such as nursing homes, the clinical management of older adults with T2DM is particularly challenging [5]. Residents are often affected by advanced age, polypharmacy, geriatric syndromes, and limited functional reserves [6–9]. In this context, the identification of reliable prognostic markers of major health events is essential to guide individualized and appropriate care strategies in this vulnerable population.

Frailty has emerged as a key determinant of clinical vulnerability in older adults, especially among those living with T2DM [10]. It is defined as a multidimensional syndrome reflecting increased vulnerability to stressors due to cumulative declines across multiple physiological systems [11]. Frailty is highly prevalent among older adults with T2DM [2,12]. The Frailty Index (FI), based on the accumulation of health deficits, is currently one of the most widely used and validated tools for identifying frailty. These deficits encompass a broad spectrum of variables, including clinical signs, symptoms, chronic diseases, functional impairments, psychosocial risk factors, and common geriatric syndromes. The FI is calculated as a ratio of the number of deficits present to the total number of considered variables, thereby providing a continuous score that reflects an individual's biological vulnerability. Unlike categorical approaches to frailty, the FI captures the multidimensional and progressive nature of the aging process. It has consistently demonstrated strong predictive validity for a range of adverse health outcomes, such as hospitalizations, admission to long-term care facilities, functional decline, and all-cause mortality, particularly among community-dwelling older adults [6,13–17]. Importantly, the FI may serve as a valuable tool to distinguish between chronological age and biological age, offering a more nuanced understanding of heterogeneity in aging trajectories beyond specific diagnostic categories. This capacity to stratify risk at the individual level underscores its potential utility for tailoring prevention strategies, care planning, and resource

allocation in geriatric populations. In the French Caribbean, the prevalence of T2DM is higher than in mainland France. Recent findings from nursing homes in the French West Indies [18] report T2DM rates exceeding 28%, well above the national average. In this area, older adults with T2DM residing in French Caribbean nursing homes exhibit a high prevalence of cardiovascular risk factors and are at risk of overtreatment. Despite this high prevalence, little is known about the prognosis value of frailty, as measured by the FI, in predicting mortality among institutionalized older adults with T2DM.

In this study, we hypothesize that the FI may serve as a useful tool for identifying the most vulnerable individuals—those at greatest risk of negative outcomes, including death—even among a population already characterized by advanced age and chronic illness. The objective of this study is to investigate the association between the FI and 12-month all-cause mortality among older adults with T2DM residing in nursing homes in the French Caribbean.

## Methods

### Study design

The KASEHPAD study was a prospective, observational study conducted in six nursing homes located in Martinique and Guadeloupe (French West Indies). This study aimed to describe the one-year health trajectories of older adults residing in nursing homes. Details regarding the inclusion criteria and baseline characteristics have been published previously [7]. In brief, participants were interviewed on-site at baseline, after 6 months, and after 12 months. At each visit, various geriatric domains were assessed, including dependency, cognition, malnutrition, neuropsychiatric symptoms, and quality of life. Treatments and comorbidities were extracted from medical records. In addition, phone interviews were conducted twice— at 3 and 9 months after baseline—to collect follow-up data. These phone calls provided updates on vital status and healthcare service utilization (i.e., types and frequency of use). In the event of a participant's death, the date and cause of death were sought and documented. The study received ethical approval from the EST 1 French Ethics Committee on June 2, 2020, and was registered on ClinicalTrials.gov on October 13, 2020 (NCT04587466). Recruitment of participants began on September 01, 2020 and the end of follow-up was November 30, 2023. The KASEHPAD study is an observational research project involving human participants, with no identified risks to participant safety. In line with this classification, the requirement for written consent was waived by the EST 1 French Ethics Committee (Reference: 2020-A00960-39), in accordance with French regulatory Law No. 2012−300. Participants received a written information leaflet outlining the key elements of the study and provided verbal consent to participate. Verbal consent was documented in the case report form. Participation was entirely voluntary, and individuals were free to decline or withdraw from the study at any time, without any negative consequences. Given the high prevalence of cognitive impairment among nursing home residents, the on-site investigator ensured that participants fully understood the implications of their involvement—namely, responding to medical questionnaires without any impact on their medical care—while explaining the study details and participants' rights, particularly regarding personal data protection. In instances where a participant was under legal guardianship and/ or unable to comprehend the study, the information leaflet was provided to the legal guardian or a designated contact person, and verbal consent was obtained and recorded in the case report form. All procedures were carried out in accordance with relevant ethical guidelines and regulatory requirements.

### Frailty index construction

For the present analysis, we constructed a frailty index in accordance with the standard procedure described by Searle and Rockwood [14,19,20]. The FI was derived from baseline data and included 30 variables encompassing a broad range of health domains, including comorbidities, cognitive and psychological status, functional capacity, and clinical signs observed during physical and neurological examinations (Table 1). Each variable (or "deficit") was dichotomized as 0 (absence of the deficit) or 1 (presence of the deficit). In total, 30 variables were considered for the computation of the FI, thereby ensuring that our model is sufficiently robust [14,19,20]. The FI score was calculated for each participant as the ratio of the number of deficits present to the total number of deficits evaluated. For example, a participant with six deficits

**Table 1. List of variables used to construct the 30-Item KASEHPAD FI. SPPB: Short Physical Performance Battery, MNA: Mini Nutritional Assessment.**

| |
|---|
| 1. Atrial fibrillation |
| 2. Arterial Hypertension |
| 3. Coronary heart disease |
| 4. Congestive heart failure |
| 5. Depression |
| 6. Osteoarthritis |
| 7. Osteoporosis |
| 8. Respiratory disease |
| 9. Lung problems |
| 10. Kidney disease |
| 11. Liver disease |
| 12. Thyroid disease |
| 13. Pain |
| 14. Hearing Loss |
| 15. Decreased Visual Acuity |
| 16. Dementia |
| 17. Parkinson's disease |
| 18. Stroke |
| 19. Cancer |
| 20. Bathing |
| 21. Dressing |
| 22. Toileting |
| 23. Transferring |
| 24. Urinary incontinence |
| 25. Feeding |
| 26. SPPB gait speed |
| 27. SPPB chair stand |
| 28. SPPB balance |
| 29. Weight loss (MNA scale) |
| 30. Neuropsychological problems (MNA scale) |

https://doi.org/10. 1371/journal. pone. 0332330. t001

had an FI score of 6/30 = 0.2. Thus, FI scores ranged from 0 (no deficits) to 1 (all deficits present), with each additional deficit increasing the score by approximately 0.030.

## Outcomes

Participants were followed for a 12 months period. Follow-up assessments included two in-person visits at 6 and 12 months, as well as two telephone interviews conducted at 3- and 9-months post-baseline. The primary outcome was all-cause mortality over the 12-month follow-up period.

## Other Variables

Sociodemographic data included age, sex, and education level. Clinical data, including medical diagnoses and current medications, were extracted from participants' medical records and healthcare documentation. Global cognitive function

was assessed using the 30-item Mini-Mental State Examination (MMSE) [21]. Physical function were evaluated using the Activities of Daily Living (ADL) score [22], while nutritional status was measured using the short form of the Mini Nutritional Assessment [23].

## Statistical Analysis

Quantitative variables were expressed as means ± standard deviations (SD), and categorical variables were presented as frequencies and percentages. Group comparisons were performed using chi-square tests for categorical variables and Student's t-tests for continuous variables. The primary outcome was 12-month all-cause mortality, treated as a binary dependent variable. To assess the association between FI and mortality, logistic regression models were employed. Both unadjusted and adjusted models were estimated, with adjustments made for age and sex. The strength of association was expressed as odds ratios (ORs) with 95% confidence intervals (CIs). Additionally, Cox proportional hazard models were performed to study the relationship between the FI and the risk of mortality over the follow-up. Missing data were not imputed. All statistical analyses were conducted using R software (R Foundation for Statistical Computing, Vienna, Austria).

## Result

In total, 94 older adults with T2DM were included in the study and 91 were analyzed at 12 months follow-up (Fig 1). The mean age of participants was 81.1 ± 10.0 years, and 42.6% were male. During the 12-month follow-up period, 28 (29.8%) older adults died. Compared to survivors, those who died were significantly older (88.0 ± 9.9 vs 78.2 ± 8.6 years; $p < 0.001$) and had a lower body mass index (20.8 ± 4.5 vs 25.5 ± 5.2; $p < 0.001$). Dementia was more frequent among those who died (75.0% vs 44.4%; $p = 0.007$), and they exhibited significantly poorer functional and cognitive performance, as reflected by lower ADL score (1.3 ± 1.6 vs 2.5 ± 2.1; $p = 0.003$) and MMSE score (6.7 ± 6.5 vs 12.3 ± 9.3; $p = 0.002$). The FI was not significantly higher in participants who died compared to those who survived (0.37 ± 0.13 vs 0.27 ± 0.14; $p = 0.346$). (Table 2).

In unadjusted logistic regression models, each 0.01-point increase in the FI was associated with a 6% increase in the odds of death (OR = 1.06; 95% CI: 1.02–1.11; $p = 0.002$). After adjusting for age, sex, the association remained marginally significant (adjusted OR = 1.05; 95% CI: 1.00–1.10; $p = 0.056$) (Table 3). In unadjusted Cox proportional hazard models, FI was also associated with increased risk of mortality (HR:1.04, 95%CI: 1.01–1.11; p = 0.009). After adjusting for age and sex, this association was not significant (HR: 1.03, 95%CI: 0.99–1.07; p = 0.139).

## Discussion

In this prospective cohort study of older adults with T2DM residing in nursing homes in the French Caribbean, our findings suggest that FI mays serve as a predictor of 12-month all-cause mortality. Individuals with higher frailty levels were significantly more likely to die within one year in univariates analysis. In multivariable models adjusted for age and gender, frailty was marginally associated with 1-year mortality when using logistic regression but was not significantly associated with time to death in the Cox proportional hazards model. This discrepancy likely reflect the different nature of the models, with logistic regression capturing the overall likelihood of death at one year, and Cox regression focusing on the instantaneous hazard over time. conceptual frameworks of the two models: logistic regression estimates the overall probability of death within a fixed period, whereas Cox regression evaluates the instantaneous hazard of death over time. While we hypothesized that frailty would be associated with increased overall mortality among residents with T2DM, it may not significantly influence the timing of death over the 12-month follow-up period. Nevertheless, this observation warrants further validation through large-scale studies.

Our results are consistent with previous studies of the relationship between FI and adverse health outcomes in older adults [24,25]. These studies have demonstrated that FI is a strong predictor of death [6,26], functional decline [27], and hospitalization in both community-dwelling and institutionalized older adults [28]. However, the strength of this association

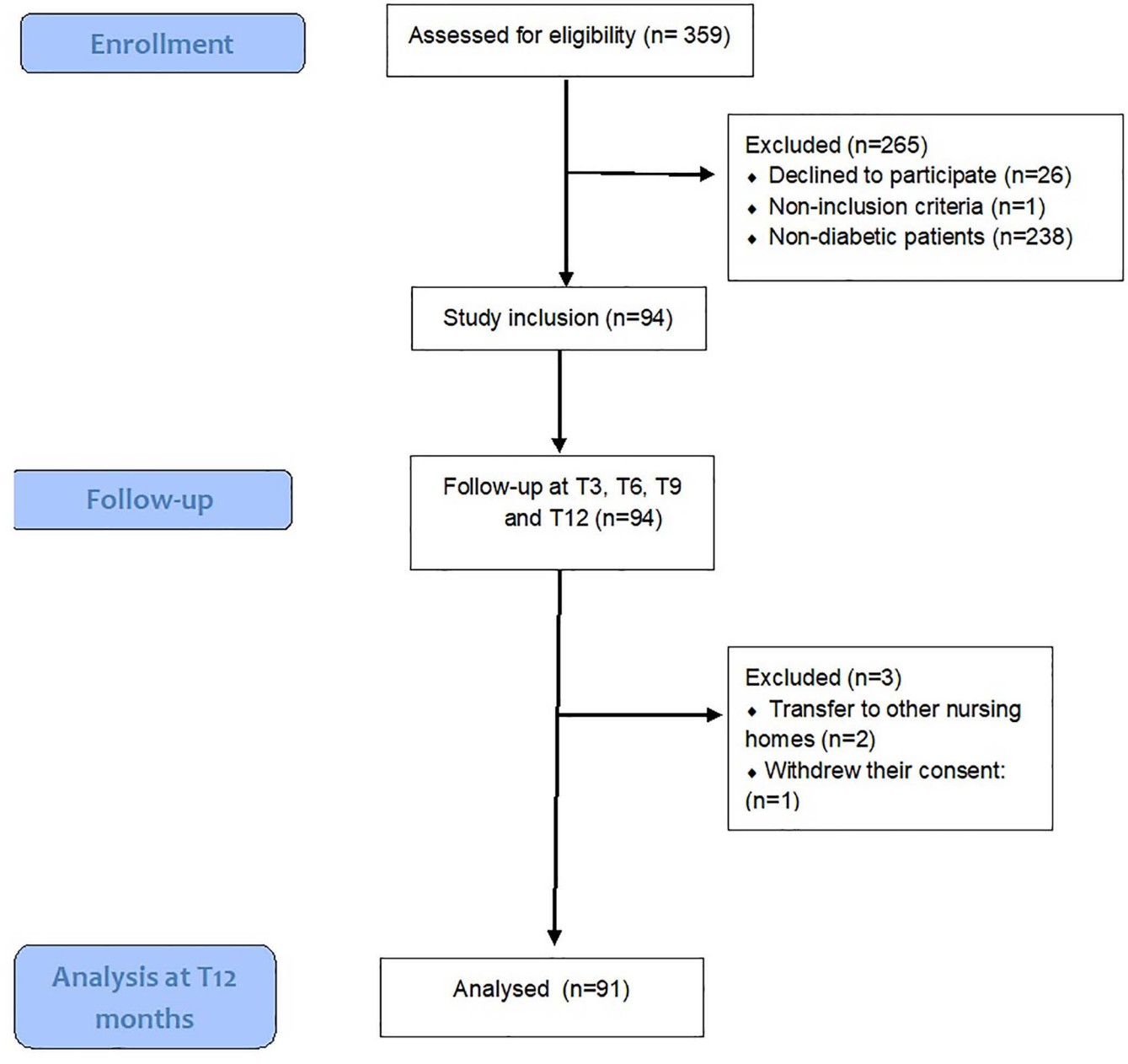

**Fig 1. Flow-chart of the KASEHPAD study regarding older adults with T2DM.**

may be attenuated in nursing home populations, likely due to the high baseline prevalence of deficits and proximity to end-of-life. This "ceiling effect" suggests a reduced discriminatory capacity of the FI in highly frail populations, as previously reported by Tabue-Teguo and colleagues [6]. Importantly, while frailty has been widely studied in general geriatric cohorts, few investigations have specifically targeted diabetic nursing home residents—a group characterized by advanced age, high levels of multimorbidity, and complex clinical needs [12,29,30]. Our study addresses this gap and provides evidence that frailty, as measured by the FI, could be a clinically meaningful predictor of short-term mortality in this high-risk

**Table 2. Baseline Characteristics of Nursing Home Residents According to Death Events (n = 94).**

| Characteristics | Mean ± CI or n (%) | Death Event (=yes) (n = 28) | Death Event (=no) (n = 63) | p |
|---|---|---|---|---|
| Age | 81.1 ± 10.0 | 88.0 ± 9.9 | 78.2 ± 8.6 | <0.001 |
| Gender (men) | 40 (42.6%) | 10 (35.7%) | 30 (47.6%) | 0.291 |
| BMI | 24.4 ± 5.3 | 20.8 ± 4.5 | 25.5 ± 5.2 | <0.001 |
| Hypertension | 80 (85.1%) | 24 (85.7%) | 54 (85.7%) | 1 |
| Cardiac failure | 16 (17.0%) | 4 (14.3%) | 12 (19.0%) | 0.768 |
| Myocardial infarction | 5 (5.3%) | 2 (7.1%) | 3 (4.5%) | 0.641 |
| Stroke | 23 (24.5%) | 6 (21.4%) | 17 (27.0%) | 0.573 |
| Dementia | 50(53.2%) | 21 (75.0%) | 28 (44.4%) | 0.007 |
| Parkinson's disease | 6 (6.4%) | 4 (14.3%) | 2 (3.2%) | 0.070 |
| Depression | 15 (16.0%) | 4 (14.3%) | 10 (15.9%) | 1 |
| Kidney disease | 21 (22.3%) | 4 (14.3%) | 17 (27.0%) | 0.184 |
| ADL score | 2.1 ± 2.0 | 1.3 ± 1.6 | 2.5 ± 2.1 | 0.003 |
| MMSE score | 10.4 ± 8.9 | 6.7 ± 6.5 | 12.3 ± 9.3 | 0.002 |
| FI score | 0.30 ± 0.14 | 0.37 ± 0.13 | 0.27 ± 0.14 | 0.346 |

BMI: Body Mass Index; ADL: Activities of Daily Living; MMSE: Mini-Mental State Examination; FI: Frailty Index

**Table 3. Relationship of Frailty Index (FI) and Mortality over 1 year of follow-up.**

| | Logistic model | | | | Cox model | | | |
|---|---|---|---|---|---|---|---|---|
| | Unadjusted OR | p | Adjusted OR | p | Unadjusted HR | p | Adjusted HR | p |
| FI | 1.06 (1.02-1.11) | 0.002 | 1.05 (1.00-1.10) | 0.056 | 1.04 (1.01-1.11) | 0.009 | 1.03 (0.99-1.07) | 0.139 |

OR: odd-Ratio. HR: Hazard Ratio

subgroup. The observed 12-month mortality rate of nearly 30% among frail residents with T2DM is concerning but consistent with their known vulnerability. Frailty likely reflects the cumulative impact of physiological decline across multiple systems, impairing resilience to common acute stressors such as infections, cardiovascular events, or exacerbations of chronic diseases [31,32]. The FI, as a quantitative and scalable tool, may offer clinicians a practical approach to stratifying risk and guiding care decisions, including advance care planning and prioritization of interventions [33]. Notably, in our analysis, the FI outperformed several individual comorbidities (e.g., hypertension, chronic kidney disease) in predicting mortality, emphasizing the added prognostic value of comprehensive geriatric assessment over disease-specific indicators alone. Although longitudinal assessment of FI could be of interest, the relatively short 12-month follow-up period in our study limits the potential added value of measuring changes over time. Previous research using the same set of FI variables [6] demonstrated that the baseline FI reliably predicted 1-year mortality in nursing home residents. Given the deficit accumulation approach used to construct the FI [34], it tends to remain stable over short intervals, which may reduce the utility of repeated assessments within a 6-month timeframe. Furthermore, our relatively small sample size (94 participants with T2DM) and the limited number of events (28 deaths) reduce the statistical power to detect additional predictive value from mid-point FI measurements.

This study has several strengths, including a longitudinal design, the use of validated frailty assessment methodology, and a focus on an underrepresented and particularly vulnerable population. From a clinical standpoint, findings from the KASEHPAD cohort offer valuable real-world insights into the prognostic significance of frailty in diabetic older adults living in long-term care facilities in the French West Indies. Nonetheless, some limitations should be acknowledged. The relatively small sample size may have limited the statistical power to detect subgroup differences or explore interactions.

Notably, a more detailed analysis of macrovascular and microvascular complications and treatments could provide valuable insights into the relationship between these complications and mortality. The magnitude of the association between frailty and mortality was low. This association is likely to be weaker in nursing home populations because residents generally present with a high burden of health deficits and are, therefore, closer to the outcome of interest—namely, mortality—compared to community-dwelling older adults. Moreover, while the FI was constructed according to established principles, it was adapted to the available dataset, which could affect its reproducibility. Finally, the absence of cause-specific mortality data limits our ability to elucidate the specific pathways linking frailty and death.

## Conclusion

In this study, frailty measured by the FI showed a tendency to be associated with short-term mortality among older adults with type T2DM living in nursing homes. While these findings did not reach strong statistical significance, the observed trend suggests that frailty may play an important prognostic role in this vulnerable population. Incorporating frailty assessment into routine care may contribute to more personalized and appropriate management strategies. Nevertheless, these findings warrant larger-scale studies to further support the prognostic utility of the Frailty Index in guiding the management and care planning of institutionalized older adults with T2DM.

## Supporting information

**S1 Data. Data Set.**
(CSV)

## Acknowledgments

We would like to thank the ACTIVE Team from Bordeaux for their precious methodological support, as well as Valérie Soter, and Mélanie Petapermal for their regulatory support. We thank the following nursing homes for participating in the study: EHPAD Les Flamboyants (Gourbeyre, Guadeloupe), EHPAD Kalana (Bouillante, Guadeloupe), EHPAD Nou Grand Moun (Capesterre-Belle-Eau, Guadeloupe), EHPAD les Jardins de Belost (Saint-Claude, Guadeloupe), Centre Hospitalier Gerontologique Palais Royal (Les Abymes, Guadeloupe) and Centre Emma Ventura (Fort-de-France, Martinique).

**Declaration of generative AI:** During the preparation of this work the author(s) used ChatGPT in order to improve readability and language. After using this tool/service, the author(s) reviewed and edited the content as needed and take(s) full responsibility for the content of the publication.

## Author contributions

**Conceptualization:** Maturin Tabué-Teguo, Denis Boucaud-Maitre.

**Formal analysis:** Denis Boucaud-Maitre.

**Funding acquisition:** Maturin Tabué-Teguo, Moustapha Dramé.

**Investigation:** Maturin Tabué-Teguo, nadine Simo, Laurys Letchimy, Denis Boucaud-Maitre.

**Methodology:** Maturin Tabué-Teguo, Moustapha Dramé, Denis Boucaud-Maitre.

**Supervision:** Maturin Tabué-Teguo, Moustapha Dramé.

**Validation:** Maturin Tabué-Teguo, Moustapha Dramé, Denis Boucaud-Maitre.

**Writing – original draft:** Maturin Tabué-Teguo, Axiane Placide-Francil.

**Writing – review & editing:** nadine Simo, Moustapha Dramé, Laurys Letchimy.

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
