## [Decision Letter · Decision Letter 0]

18 Jul 2025

PONE-D-25-31998
Frailty and 12-Month Mortality Among Older Adults with Type 2 Diabetes in Nursing Homes. A longitudinal study.
PLOS ONE

Dear Dr. Boucaud-Maitre,

Thank you for submitting your manuscript to PLOS ONE. After careful consideration, we feel that it has merit but does not fully meet PLOS ONE’s publication criteria as it currently stands. Therefore, we invite you to submit a revised version of the manuscript that addresses the points raised during the review process.

We look forward to receiving your revised manuscript.

Kind regards,

Mario Ulises Pérez-Zepeda, M.D., Ph.D.

Academic Editor

PLOS ONE

Journal Requirements:

4. Please update your submission to use the PLOS LaTeX template. The template and more information on our requirements for LaTeX submissions can be found at http://journals.plos.org/plosone/s/latex.

5. Please amend your list of authors on the manuscript to ensure that each author is linked to an affiliation. Authors’ affiliations should reflect the institution where the work was done (if authors moved subsequently, you can also list the new affiliation stating “current affiliation:….” as necessary).

6. Please remove all personal information, ensure that the data shared are in accordance with participant consent, and re-upload a fully anonymized data set.

Reviewers' comments:

Reviewer's Responses to Questions

**Comments to the Author**

1. Is the manuscript technically sound, and do the data support the conclusions?

Reviewer #1: No

Reviewer #2: No

2. Has the statistical analysis been performed appropriately and rigorously? 

Reviewer #1: No

Reviewer #2: N/A

3. Have the authors made all data underlying the findings in their manuscript fully available?

Reviewer #1: Yes

Reviewer #2: No

4. Is the manuscript presented in an intelligible fashion and written in standard English?

Reviewer #1: Yes

Reviewer #2: Yes

5. Review Comments to the Author

Reviewer #1: Considering that this is a mortality study, the authors should have done a survival analysis including log rank test and Cox regression. They seem to have the data, so in my opinion they should perform it in order to publish their results.

Reviewer #2: Thank you for the opportunity to review this manuscript addressing the prognostic role of frailty in older adults with type 2 diabetes mellitus (T2DM) residing in nursing homes. The topic is highly relevant given the growing interest in frailty and risk stratification in institutionalized older adults with multimorbidity. However, I believe that the manuscript requires substantial revision before it can be considered for publication. Below, I provide detailed comments intended to help the authors strengthen their work.

Major Comments:

1.Lack of Stratification by Diabetes Treatment and Complications

The manuscript does not distinguish between participants treated with insulin and those on oral antidiabetic medications, nor does it stratify for the presence or absence of macrovascular complications. This is a significant limitation, as the severity and management of diabetes (including history of cardiovascular events, renal impairment, or diabetic foot) are likely to influence both frailty and mortality.

2.Absence of a Comparison Group Without Diabetes

The study would have benefitted from including a comparator group of non-diabetic nursing home residents to evaluate whether frailty carries a different prognostic weight in diabetic versus non-diabetic older adults. Without this comparison, it is difficult to assess the specificity of the observed associations to the diabetic population.

3.Unclear Follow-Up Design and Methodology

Although the manuscript mentions follow-up through in-person and telephone interviews, it does not detail the content or structure of these follow-up interactions—particularly the telephone assessments. Clarifying what clinical, functional, or survival data were captured at each time point (3, 6, 9, and 12 months) is crucial for evaluating the robustness of the outcome ascertainment.

4.Frailty Index Temporal Dynamics Not Reported

5.Overlap Between FI and ADL Measures

Several items included in the FI (e.g., bathing, dressing, toileting) are also captured separately using the ADL scale. The potential redundancy and statistical collinearity should be addressed. It would be important to clarify whether the predictive value of the FI is independent of the ADL score.

6.Inconsistency Between Results and Conclusions

In the results section, the adjusted analysis shows that FI is not a statistically significant predictor of mortality (p = 0.056), and yet the conclusions emphasize its utility for care planning and management. This discrepancy should be reconciled. A more nuanced interpretation is required, especially given the marginal statistical significance and the possible confounding role of age.

7.Impact on Clinical Management Not Substantiated

The authors suggest that FI assessment may guide care planning and improve prognosis, but they do not explain what specific changes in management would follow FI stratification, nor do they provide any evidence that such changes would alter outcomes. In particular, the authors should clarify how they envision the FI influencing decision-making in patients whose outcomes are predominantly driven by age and baseline functional impairment.

8.Definition of Frailty Needs Clarification

The manuscript adopts the deficit accumulation approach, but it does not adequately define or justify the conceptual framework of frailty being used. A clearer explanation of how frailty is operationalized and how it differs from disability or comorbidity would benefit readers unfamiliar with the field.

9.Underdeveloped Discussion and Conclusion

The discussion could be strengthened by a more critical interpretation of the limitations, especially regarding the sample size, generalizability, and the ceiling effect in highly frail populations. The conclusions would also benefit from being more concise and aligned with the actual findings. Currently, they overstate the prognostic utility of FI despite the lack of a significant adjusted effect and the absence of intervention data.

Additional Suggestions

Consider including a flow diagram of participant recruitment and follow-up.

Indicate how missing data were handled.

Ensure statistical results are consistently reported (e.g., precise p-values and confidence intervals).

Conclusion

In its current form, the manuscript raises important questions but does not provide sufficient methodological or interpretive clarity to support its conclusions. I recommend major revision, with careful attention to the points outlined above. I appreciate the authors’ effort in addressing a timely and clinically significant topic and hope these comments are helpful in improving the manuscript.

6. PLOS authors have the option to publish the peer review history of their article (what does this mean?). If published, this will include your full peer review and any attached files.

Reviewer #1: No

Reviewer #2: No

---

## [Author Response · Author response to Decision Letter 1]

18 Aug 2025

Reviewer #1:

1. Considering that this is a mortality study, the authors should have done a survival analysis including log rank test and Cox regression. They seem to have the data, so in my opinion they should perform it in order to publish their results.

Authors response : We thank the reviewer for their comments. First of all, we would like to clarify our decision to use a logistic regression model rather than a Cox proportional hazards model to analyze one-year mortality. Given the short and clearly defined follow-up period, the relatively low sample size, with vital status assessed precisely at 12 months, we considered logistic regression to be an appropriate and interpretable approach. Indeed, the study population consisted of older adults living in nursing homes, a setting associated with high short-term mortality (approximately 30%). In this context, the proportional hazards assumption required for Cox modeling may not have been appropriate, and analyzing mortality as a binary outcome at a fixed time point provided a clearer interpretation of the event of interest. If frail patients die at approximately the same rate as non-frail patients over time — that is, if their survival curves are similar — the Cox proportional hazards model may not detect a significant association, even if the overall mortality is slightly higher in the frail group. This is because the Cox model estimates the hazard ratio, which reflects the instantaneous risk of death at any given time point, rather than the cumulative probability of death over the entire follow-up period. Therefore, if the excess mortality among frail individuals is spread evenly over time without a clear difference in the timing of deaths, the hazard ratio may be close to 1 and non-significant, despite a potentially meaningful difference in overall mortality. In contrast, logistic regression — which compares the final mortality status at a fixed time point (e.g., 1 year) — may be more sensitive to such cumulative effects, particularly in studies with relatively short follow-up periods as observed in KASEHPAD study or when most events occur late during follow-up (more than half of death occur between six and twelwe months in our study). Therefore, we considered the logistic model to be more appropriate for addressing our research question.

Nevertheless, we acknowledge that a Cox model can provide complementary insights, and we have performed this additional analysis. Using the Cox model (n = 91), we found that higher FI scores were significantly associated with increased mortality risk: each 0.01-point increase in the FI was associated with a 4.3% increase in the hazard of death (HR = 1.04; 95% CI: 1.01–1.11; p = 0.0089). This result is similar to those obtained using logistic regression. However, after adjustment for age and sex, the association between FI and mortality was no longer significant (adjusted HR = 1.03; 95% CI: 0.99–1.07; p = 0.139), while age remained significantly associated with mortality (HR = 1.08; 95% CI: 1.03–1.13; p < 0.001). This suggests that while frailty may be associated with increased overall mortality, it may not significantly influence the timing of death during the 12 months follow-up.

We propose to include the Cox analysis in the Methods and Results sections, and to add the following paragraph to the Discussion: “ In multivariable models adjusted for age and gender, frailty was marginally associated with 1-year mortality when using logistic regression but was not significantly associated with time to death in the Cox proportional hazards model. This discrepancy likely reflect the different nature of the models, with logistic regression capturing the overall likelihood of death at one year, and Cox regression focusing on the instantaneous hazard over time. conceptual frameworks of the two models: logistic regression estimates the overall probability of death within a fixed period, whereas Cox regression evaluates the instantaneous hazard of death over time. While we hypothesized that frailty would be associated with increased overall mortality among residents with T2DM, it may not significantly influence the timing of death over the 12-month follow-up period. Nevertheless, this finding warrants confirmation through large-scale studies.”

Reviewer #2: Thank you for the opportunity to review this manuscript addressing the prognostic role of frailty in older adults with type 2 diabetes mellitus (T2DM) residing in nursing homes. The topic is highly relevant given the growing interest in frailty and risk stratification in institutionalized older adults with multimorbidity. However, I believe that the manuscript requires substantial revision before it can be considered for publication. Below, I provide detailed comments intended to help the authors strengthen their work.

Authors comment : We thank the reviewer for their encouraging comments.

Major comments

1. Lack of Stratification by Diabetes Treatment and Complications

The manuscript does not distinguish between participants treated with insulin and those on oral antidiabetic medications, nor does it stratify for the presence or absence of macrovascular complications. This is a significant limitation, as the severity and management of diabetes (including history of cardiovascular events, renal impairment, or diabetic foot) are likely to influence both frailty and mortality.

Authors response : We concur with the reviewer’s comment, which should be considered alongside a more descriptive study of KASEHPAD of older adults residing in nursing homes that we recently conducted and published (Tabué-Teguo M, Simo N, Rambhojan C, Letchimy L, Bonnet M, Vélayoudom FL, Boucaud-Maitre D. Prevalence and characteristics of older adults with type 2 diabetes mellitus living in French Caribbean nursing homes: results from the baseline KASEHPAD study. Aging Clin Exp Res. 2025 Mar 24;37(1):103. doi: 10.1007/s40520-025-03008-5. PMID: 40128462; PMCID: PMC11933193). In this study, the mean HbA1c was 7.32% ± 1.5%, with 35 participants (42.7%) exhibiting an HbA1c level of <7%. Among the residents, 37.2% were not receiving any antidiabetic treatment, 43% were on insulin (n=41), and 25% were receiving oral antidiabetic agents.

However, we acknowledge that the sample size may be too small to perform robust additional analyses. Moreover, we observed in our study that, in multivariate analysis (n = 71) adjusted for age, sex, BMI, hypertension, hypercholesterolemia, dementia, MNA score, ADL score, and HbA1c, only HbA1c was significantly associated with antidiabetic treatment (OR: 1.76; 95% CI: 1.12–3.04). This suggests that antidiabetic treatment may not be a reliable marker of clinical severity when comorbidities are taken into account.

Regarding comorbidities, we used a frailty index based on the model proposed by Rockwood, which includes atrial fibrillation, hypertension, coronary heart disease, heart failure, and kidney disease. We agree with the reviewer that a more detailed analysis of macrovascular and microvascular complications could provide valuable insights into the relationship between these complications and mortality. We acknowledge this as a major limitation of our study and propose to explicitly address it in the discussion section.

2. Absence of a Comparison Group Without Diabetes.

The study would have benefitted from including a comparator group of non-diabetic nursing home residents to evaluate whether frailty carries a different prognostic weight in diabetic versus non-diabetic older adults. Without this comparison, it is difficult to assess the specificity of the observed associations to the diabetic population.

Authors response: This is an interesting point. As the KASEHPAD cohort also included non-diabetic participants (n=224), we conducted an additional analysis to examine the association between the frailty index and one-year mortality in non-diabetic residents—a relationship that has already been demonstrated in previous studies. In our logistic regression model (n=208), the frailty index was indeed associated with one-year mortality: OR = 1.06 (95% CI: 1.03–1.09), as was male sex (OR = 2.24; 95% CI: 1.05–4.95), while age was not significantly associated (OR = 1.03; 95% CI: 0.99–1.07).

This additional analysis reinforces the external validity of our study, as it confirms findings already reported in the literature. However, we would like to emphasize that this analysis among non-diabetic individuals lies outside the primary scope of our study focused on older adults with T2DM residing in nursing homes. Given that this association has been well-documented previously, we do not consider it necessary to include these additional findings in the main manuscript.

3.Unclear Follow-Up Design and Methodology. Although the manuscript mentions follow-up through in-person and telephone interviews, it does not detail the content or structure of these follow-up interactions—particularly the telephone assessments. Clarifying what clinical, functional, or survival data were captured at each time point (3, 6, 9, and 12 months) is crucial for evaluating the robustness of the outcome ascertainment.

Author’s response : We agree with the reviewer’s comment. The methodology of this study has already been described in several publications (Boucaud-Maitre D et al., Front Med (Lausanne). 2024 Sep 17;11:1428443; Boucaud-Maitre D et al., Sci Rep. 2025 Feb 20;15(1):6170; Boucaud-Maitre D et al., Sci Rep. 2025 Mar 6;15(1):7918). Nevertheless, we acknowledge the importance of providing sufficient methodological detail for readers of the present article. We therefore propose to expand the Methods section as follows:

“The KASEHPAD study was a prospective, observational study conducted in six nursing homes located in Martinique and Guadeloupe (French West Indies). The aim of the study was to describe the one-year health trajectories of older adults residing in nursing homes. Details regarding the inclusion criteria and baseline characteristics have been published previously (7). In brief, participants were interviewed on-site at baseline, after 6 months, and after 12 months. At each visit, various geriatric domains were assessed, including dependency, cognition, malnutrition, neuropsychiatric symptoms, and quality of life. Treatments and comorbidities were extracted from medical records. In addition, phone interviews were conducted twice—at 3 and 9 months after baseline—to collect follow-up data. These phone calls provided updates on vital status and healthcare service utilization (i.e., types and frequency of use). In the event of a participant’s death, the date and cause of death were sought and documented.”

4. Frailty Index Temporal Dynamics Not Reported.

Author’s response : We thank the reviewer for this pertinent comment. We agree that analyzing the evolution of the Frailty Index (FI) over time is of interest. However, in our study, the 6-month follow-up period is relatively short and does not provide additional relevant information regarding the association between FI and mortality.

The Rockwood FI measured at baseline has been well established in the literature as a robust predictor of short- and medium-term mortality (Mitnitski et al., 2001; Rockwood et al., 2005) even to nursing home. In a previous study (Tabue Teguo et al., JAMDA 2015), we also demonstrated that the baseline FI constructed using the same variables as in the present work accurately predicted 1-year mortality in nursing home residents. Its short-term stability (over 6 months), partly due to its deficit accumulation based methodology (Searle et al.), limits the added value of 6 months measurements. Finally, given the sample size (90 T2DM participants) and the relatively low number of events (30 deaths) in our study, including the 6-month FI measurement would be unlikely to improve the predictive performance or provide clinically meaningful additional insights.We have added a corresponding paragraph in the discussion section.

5. Overlap Between FI and ADL Measures. Several items included in the FI (e.g., bathing, dressing, toileting) are also captured separately using the ADL scale. The potential redundancy and statistical collinearity should be addressed. It would be important to clarify whether the predictive value of the FI is independent of the ADL score.

Author’s response : There was no collinearity in our multivariate analysis, as we adjusted for age and sex, but not for ADL, which are already included in the frailty index.

6. Inconsistency Between Results and Conclusions. In the results section, the adjusted analysis shows that FI is not a statistically significant predictor of mortality (p = 0.056), and yet the conclusions emphasize its utility for care planning and management. This discrepancy should be reconciled. A more nuanced interpretation is required, especially given the marginal statistical significance and the possible confounding role of age.

Author’s response : We fully agree with the reviewer’s objection. Our study raises a scientifically relevant question that deserves to be explored in larger-scale studies, in order to determine whether the assessment of frailty in nursing home settings is indeed essential. We have revised both the abstract, the discussion and the conclusion to reflect this more nuanced interpretation.

7. Impact on Clinical Management Not Substantiated

The authors suggest that FI assessment may guide care planning and improve prognosis, but they do not explain what specific changes in management would follow FI stratification, nor do they provide any evidence that such changes would alter outcomes. In particular, the authors should clarify how they envision the FI influencing decision-making in patients whose outcomes are predominantly driven by age and baseline functional impairment.

Author’s response: We agree with the reviewer. At this stage, it is necessary to confirm (or refute) these findings before recommending the widespread implementation of the Frailty Index in nursing homes for T2DM adults. See discussion and conclusion.

8. Definition of Frailty Needs Clarification. The manuscript adopts the deficit accumulation approach, but it does not adequately define or justify the conceptual framework of frailty being used. A clearer explanation of how frailty is operationalized and how it differs from disability or comorbidity would benefit readers unfamiliar with the field.

Author’s response : We agree with the reviewer’s comment. The Frailty Index (FI), developed by Rockwood and colleagues, is grounded in a theoretical framework that conceptualizes frailty as the result of an age-related accumulation of health deficits. We propose to develop this point in the introduction section : « These deficits encompass a broad spectrum of variables, including clinical signs, symptoms, chronic diseases, functional impairments, psychosocial risk factors, and common geriatric syndromes. The FI is calculated as a ratio of the number of deficits present to the total number of considered variables, thereby providing a continuous score that reflects an individual's biological vulnerability. Unlike categorical approaches to frailty, the FI captures the multidimensional and progressive nature of the aging process. It has consistently demonstrated strong predictive validity for a range of adverse health outcomes, such as hospitalizations, admission to long-term care facilities, functional decline, and all-cause mortality, particularly among community-dwelling older adults. Importantly, the FI may serve as a valuable tool to distinguish between chronological age and biological age, offering a more nuanced understanding of heterogeneity in aging trajectories beyond specific diagnostic categories. This capacity to stratify risk at the individual level underscores its potential utility for tailoring prevention strategies, care planning, and resource allocation in geriatric populations. »

9. Underdeveloped Discussion and Conclusion. The discussion could be strengthened by a more critical interpretation of the limitations, especially regarding the sample size, generalizability, and the ceiling effect in highly frail populations. The conclusions would also benefit from being more concise and aligned with the actual findings. Currently, t

---

## [Editor Report · Decision Letter 1]

29 Aug 2025

Frailty and 12-Month Mortality Among Older Adults with Type 2 Diabetes in Nursing Homes. A longitudinal study.

PONE-D-25-31998R1

Dear Dr. Boucaud-Maitre,

We’re pleased to inform you that your manuscript has been judged scientifically suitable for publication and will be formally accepted for publication once it meets all outstanding technical requirements.

Kind regards,

Mario Ulises Pérez-Zepeda, M.D., Ph.D.

Academic Editor

PLOS ONE
---

## [Editor Report · Acceptance letter]

PONE-D-25-31998R1

PLOS ONE

Dear Dr. Boucaud-Maitre,

I'm pleased to inform you that your manuscript has been deemed suitable for publication in PLOS ONE. Congratulations! Your manuscript is now being handed over to our production team.

Kind regards,

on behalf of

Dr. Mario Ulises Pérez-Zepeda

Academic Editor

PLOS ONE